# Sectoral sensitivity of the Kuwait stock market to a dual shock

Talal Alotaibi[1], Lucía Morales[2,3]*

1 Technological University Dublin, Dublin, Ireland, 2 Faculty of Business, School of Accounting, Economics and Finance, Department of Economics, Technological University Dublin, Dublin, Ireland, 3 European University of Technology (EUt+)

* lucia.morales@tudublin.ie

## Abstract

This study examines Kuwait's stock market sectors' response to the 2020 dual shock of the COVID-19 pandemic and the oil price war between Russia and Saudi Arabia. Traditional cointegration and causality models support the analysis alongside the frequency domain to assess short-run dynamics. The research sample integrates daily data from December 31, 2015, to February 23, 2022. The core research findings indicate that the consumer services, industrials, and basic materials sectors were the most affected by fluctuations in oil prices, highlighting the vulnerability of Kuwait's oil-dependent economy. Furthermore, the results from the econometric modelling reveal a significant long-run relationship with the West Texas Intermediate index (WTI) and short-run dynamics with the Brent, OPEC and Dubai oil benchmarks. At the global level, monitoring the impact of shocks in commodity-driven economies can help in designing policies that minimise the effects of systemic risks, affecting energy supply chains and inflation stability. This research advances the field by providing a multi-methodological analysis integrating traditional and efficient econometric models to assess sectoral sensitivities in the context of Kuwait, which is an understudied oil-exporting economy. The research findings offer valuable insights for investors and policy-makers in managing risks associated with oil price fluctuations during times of enhanced uncertainty.

## 1. Introduction

The year 2020 was a challenging year for the world's socioeconomic system due to a dual shock that was defined by a deadly pandemic and oil price instability as a result of the war between Russia and the Kingdom of Saudi Arabia. Oil-exporting economies were significantly affected as the economic system entered a period of hibernation due to health measures that introduced major constraints through social distancing, aiming to limit human interactions. Among these exporting economies is Kuwait, which registered a significant drop in its All-share index by 25 percent on

**Editor:** Jamel Boukhatem, University of Tunis El Manar Faculty of Economic Sciences and Management of Tunis: Universite de Tunis El Manar Faculte des Sciences Economiques et de Gestion de Tunis, TUNISIA

**Data availability statement:** All relevant data are within the manuscript and its Supporting Information files.

**Funding:** The author(s) received no specific funding for this work.

**Competing interests:** The authors have declared that no competing interests exist.

March 23, 2020 [1]. According to the OPEC Annual Report [2], Kuwait ranks among the world's leading oil-exporting nations. In 2020, Kuwait was the world's eighth largest producer of crude oil, valued at approximately 2,438 million dollars per day, accounting for approximately 4 percent of the world's volume of crude oil exports. Moreover, Kuwait accounts for 11 percent of the OPEC member's value of petroleum export countries, representing approximately 35,231 million US dollars, with 6 per cent of the world's oil reserves, by 101.5 billion barrels. The oil sector makes up nearly half of Kuwait's GDP, around 95 percent of its exports, and approximately 91 percent of government revenue [3]. The country's Industrials sector accounts for 45.4 percent of its GDP and employs 25 percent of the workforce. In contrast, the manufacturing sector remains underdeveloped, contributing only 7 percent to the national GDP, as reported by the World Bank [4,5]. The reviewed literature highlights how Kuwait's economic model is significantly exposed to the oil market dynamics [6–11]. Researchers such as Al-Yahyaei et al. [12] and Hammami [13] have offered evidence that Kuwait's heavy dependence on oil-based sectors is heavily tied to the country's oil and gas reserves. For instance, basic materials and industrials are sectors that suffered significant losses resulting from the oil price collapse and the significant economic disruptions resulting from the 2020 Global Health Crisis. During this period, oil barrels were sold at historically low prices, as WTI recorded a historic low, entering negative territory with a −36$ price that was followed the next day with −6$ for the first time in oil history [3,5,14,15]. The importance of oil for Kuwait's economy is unquestionable, with a need to have insights on implications for its sectors and their stability during times of high uncertainty, which are captured in this study's leading research question and two research hypotheses as follows,

- What is the impact of the 2020 dual market shock (COVID-19 and oil prices shock) on the Kuwait stock market sectors?

Hypotheses:

1. H10: There is evidence of a potential long-run relationship between oil benchmarks and Kuwait's stock market sectors.

    H1A: There is no evidence of a potential long-run relationship between oil benchmarks and Kuwait's stock market sectors.

2. H20: There is evidence of a potential short-run relationship between oil benchmarks and Kuwait's stock market sectors.

    H2A: There is no evidence of a potential short-run relationship between oil benchmarks and Kuwait's stock market sectors.

    The research methodology integrates well-known and established econometric models such as the Johansen and Juselius Cointegration test [16], the Engle and Granger Causality test [17], the Granger Causality test [18], and the Breitung and Candelon Frequency Domain Causality test [19] models to ensure a robust framework through the implementation of different econometric models aiming to examine consistency in results. The core research findings indicate that sectoral decline was

significant, with consumer services, industrials and basic materials particularly being the most affected sectors during the dual shock period [3,5,15].

This paper has significant implications for oil-dependent economies in the GCC and Middle Eastern regions, as they are major oil producers and hold significant reserves. The implications at the global level are paramount due to the global economy's significant dependency on oil [3,5,15,20]. The dual shock of the COVID-19 pandemic and oil price war exposed critical vulnerabilities affecting mono-economies, revealing the urgent need for economic diversification and sectoral resilience [3,5]. Kuwait's experience offers insights into Middle Eastern economies and the GCC countries facing similar oil dependencies and limited economic diversification [11]. In particular, the study offers insights into how global shocks are transmitted through financial systems in oil-exporting economies, thus affecting international commodity markets, investment flows, and regional economic stability [21,22].

The novelty of this paper is threefold:

a) The study adds to the extant literature through the comprehensive analysis of the Kuwaiti stock market sectors to shed light on their performance in the context of the COVID-19 pandemic and the oil price war between Saudi Arabia and Russia. To the best of the authors' knowledge, this dual shock remains underexplored in the academic literature, with a significant lack of studies in the context of Kuwait's stock market at a sectoral level.

b) The study integrates a solid methodological framework based on traditional and well-established econometric models that enable cross-checking of the research findings and provide evidence of their value to understanding sectoral sensitivity to higher levels of uncertainty within a multiple econometric modelling framework integrating well-established cointegration and causality models.

c) The analysis helps to outline critical insights and recommendations for oil-dependent economies in the context of enhanced uncertainty and the imperative of economic diversification within the GCC and the Middle Eastern region. It draws insights into the importance of oil for the global economy.

Understanding sectoral sensitivities improves the ability of policy-makers and investors to manage risks and design adaptive strategies in the face of future global crises [23,24]. As oil price volatility continues to shape global economic performance, especially in emerging markets, this research study contribute to the literature on macro-financial linkages and policy responsiveness in interconnected economies. The remainder of the paper proceeds as follows. Section 2 presents the relevant literature; Section 3 defines the methodological research framework. The paper's main results and discussion are presented in sections 4 and 5, and finally, section 6 concludes the paper.

## 2. Insights at the global markets level

A few papers in the academic literature have focused on examining market sectors' reactions to the 2020 dual shock. For instance, Liu [25] examined the impact of COVID-19 on China's stock market at the sectoral level. The study illustrates that the energy and technology sectors were the most impacted, while consumer staples, healthcare and utilities were least affected. Likewise, He et al. [26] studied the impact of COVID-19 on stock prices across different sectors in China, finding that transport, mining, electricity, heating and environment industries were adversely affected by the pandemic, while manufacturing, information technology, education, and healthcare industries were resilient to it. The case of the US was explored by Mazur, Dang and Vega [27], who investigated the US stock market performance during the crash of March 2020 triggered by COVID-19, using the S&P 500 index. Their findings illustrate that natural gas, food, healthcare and software stocks registered high positive returns, whereas petroleum, real estate, entertainment and hospitality sectors declined.

Additionally, Fernandez [23] analysed 30 economies worldwide. His study illustrates that no sector was isolated from the effects of COVID-19. The analysis offered insights on oil, gas, coal performance, travel and leisure, aerospace and

defense, mining and metal, life insurance, banks, household goods, autos and parts, personal goods, and media firms. The research findings illustrate that oil, gas and coal have the highest negative returns (−50%), followed by travel and leisure (−40%), aerospace and defense (−40%). Sheriff [28] investigated the stock returns of Sharia-compliant firms from different industries and found that the stock return of the information technology sector performed better than the market, while stock returns of transportation, beverages, tourism, leisure and consumer services sectors performed worse than the market during COVID-19. Likewise, Kandil Goker, Eren, and Karaca [29] investigated the impact of the COVID-19 outbreak on the Borsa Istanbul (BIST) sector index returns. Their event study analyses the impact of the pandemic on 26 sectors in BIST, showing that the sectors most impacted by negative cumulative abnormal returns and the highest decline were sports, tourism, and transportation. Along the same line, Ozturk et al. [30] analysed the impact of COVID-19 on the Turkish stock market; the study signals that the most adversely affected sectors were metal products, machinery and the banking sector. Despite the substantial economic downturn, the food and beverages, retail, real estate, and investment sectors were less affected by the pandemic. Moreover, Alber & Refaat [31] studied the impact of the COVID-19 outbreak on the returns of 17 sectors in the Egyptian stock exchange. Their study revealed that most sectors impacted by the pandemic were the contracting and construction engineering, energy and support services, IT, media and communication services, shipping and transportation, and trade and distribution. On the other hand, the banking sector, food and beverages, tobacco, and healthcare sectors reacted positively. Sayed and Eledum [32] explored the short-term effect of seven COVID-19-related events on the stock returns of the Saudi financial market. They found that the sectors most negatively affected were banks, capital goods, transportation, and commercial services, whereas telecommunication services, food, and beverages were positively affected. A summary of the extant literature and the core research outcomes focused on COVID-19 is presented in Table 1 below, signalling the limited research focused on the stock market at the sectoral level and, in the specific case of Kuwait, with a dearth of studies examining the implications of the dual shock, a research gap that this research study addresses.

The reviewed literature provides evidence of the lack of studies that examine the effects of the dual shock in the context of Kuwait, and more specifically, the dearth of research studies examining the implications for sectors and their exposure to higher levels of uncertainty. For example, the study by Al-Kandari & Al-Roumi [33] investigated the impact of the COVID-19 pandemic on Kuwait's stock market sectors. Still, it did not consider the effects of the dual shock. The authors examined 10 out of 12 sector indices that represent the listed firms in the Kuwait stock market, using daily data over the period from March 28 to April 20, 2020. These findings illustrate that the pandemic positively affected three indices: banks, telecommunications, and consumer goods. Hence, the pandemic negatively impacted oil and gas, real estate, financial services, basic materials, consumer services, insurance and industrials without insights on the implications emerging from the impact of the 2020 dual shock.

## 3. Data and econometrics modelling

The research sample supporting this study focused on the main Kuwait-weighted market index, 12 weighted sectoral indices, and four crude oil benchmarks (WTI, Brent, Dubai, OPEC). Prices were retrieved from DataStream, and when required and according to the econometric model or test under consideration, prices were transformed into continuous natural logarithm returns without missing observations over the historical period available for the Kuwait index (.BKA). The data set consists of the daily closing price, representing 1,604 observations for the Kuwait stock market All Share Index and all the Kuwait stock market sectors. The reviewed literature suggests that the required sample size to implement cointegration and causality models requires a minimum of 100 observations [34,35]. However, researchers recommend working with larger samples as they are considered to be more reliable [36–43]. The analysis starts on December 31, 2015, and ends on February 23, 2022, to avoid the influence of the Russian-Ukrainian war on oil prices that started on February 24, 2022 [44,45]. Furthermore, the sample period was chosen to avoid introducing undesired noise into the sample resulting from the 2014 oil price shocks and the lagging effects of the global economic recovery from the 2007/08

**Table 1. Stock Markets Sectoral Level Analysis.**

| Authors | Country | Variables | Research Findings |
|---|---|---|---|
| Liu [25]; He et al. [26] | China | The impact of COVID-19 on the stock market at the sectoral level. | • The energy and technology sectors were the most impacted, while consumer staples, healthcare and utilities were the least affected.<br>• The transport, mining, electricity, heating, and environmental industries were adversely affected by the pandemic, while the manufacturing, information technology, education, and healthcare industries were resilient to the pandemic. |
| Mazur, Dang and Vega [27], | USA | Impact of COVID-19 on the US stock market, using the S&P 500 index. | • Natural gas, food, healthcare and software stocks registered high positive returns, whereas petroleum, real estate, entertainment and hospitality sectors declined. |
| Fernandez [23] | Global | The impact of COVID-19 on the stock market at the sectoral level. | • No sector was isolated from the effects of COVID-19, with oil, gas and coal having the highest negative returns (−50%), followed by the negative returns of travel and leisure (−40%), aerospace and defense (−40%). |
| Kandil Goker, Eren, and Karaca [29]; Ozturk et al. [30] | Turkey | The impact of the COVID-19 outbreak on the Borsa Istanbul sector index returns. | • The impact of the pandemic on 26 sectors in BIST shows that the sectors most impacted by negative cumulative abnormal returns and the highest decline were sports, tourism, and transportation.<br>• The most adversely affected sectors were metal products, machinery and the banking sector.<br>• Despite the substantial economic downturn, food and beverages, retail, real estate, and investment sectors were less affected by the pandemic |
| Alber & Refaat [31] | Egypt | The impact of the COVID-19 outbreaks on the returns of 17 sectors in the Egyptian exchange | • Contracting and construction engineering, energy and support services, IT, media and communication services, shipping and transportation, and trade and distribution were the sectors most impacted.<br>• The banking sector, food and beverages, tobacco, and healthcare sectors reacted positively. |
| Sayed and Eledum [32] | Saudi Arabia | Analyse the short-term effect of seven COVID-19-related events on the stock returns of the Saudi financial market. | • The most negatively affected sectors were banks, capital goods, transportation, and commercial services.<br>• The telecommunication services, food, and beverages were positively affected. |

Source: Authors' elaboration (2025).

Global Economic and Financial Crisis (GEFC). These shocks resulted in extreme volatility in global oil markets and profoundly impacted oil-dependent economies such as Kuwait, as noted by Masood et al. [46], Cause [47], and Fattouh [48]. Including the data before 2015 could lead to misrepresentation of the issues under consideration by conflating the lagged effects of earlier shocks with the 2020 dual shock (COVID-19 and the oil price war) under examination. The selection of the Kuwait-weighted market index, sectoral indices and crude oil benchmarks (Brent, Dubai, OPEC and WTI) enables an in-depth analysis of sectoral sensitivity resulting from the dual shock. The details for the Kuwait All Share Index and the sectors are outlined in Table 2 below:

### 3.1. Kuwait stock market outlook

The effects of the pandemic on the Kuwait Stock Market (All Share index) are presented in Fig 1 below highlighting that March 23, 2020, recorded the lowest point across markets, which showed a negative 25 percent.

In the context of Kuwait's stock market sectors, Fig 2 illustrates that March 23, 2020, recorded the lowest point across markets. The consumer services (consumer services), the industrials index (industrials), and the basic materials index (basic materials) were the most affected, dropping by approximately 33 per cent, 32 per cent, and 31 per cent, respectively, which was higher than the All-Share index. The banks index (banks), the financial services index (financial services) and the real-estate index (real estate) registered significant drops with 28 percent, 24 percent and 23 percent, respectively.

**Table 2. Data Set.**

| Benchmark | Companies |
|---|---|
| All Share Index | All Share Index (171 Companies) |
| Bank Sector | Bank (11 companies) |
| Real Estate Sector | Real Estate (39 Companies) |
| Financial Services Sector | Financial Services (49 Companies) |
| Telecommunications Sector | Telecom (5 Companies) |
| Consumer Services Sector | Consumer Services (13 Companies) |
| Oil & Gas Sector | Oil & Gas (6 Companies) |
| Healthcare Sector | Healthcare (3 Companies) |
| Insurance Sector | Insurance (9 Companies) |
| Basic Materials Sector | Basic Materials (4 Companies) |
| Consumer Goods Sector | Consumer Goods (3 Companies) |
| Industrials Sector | Industrials (28 Companies) |
| Technology Sector | Technology (1 Company) |

Table 2: This details the Kuwait All Share index and its sectors, with Financial Services (49 companies) and Real Estate (39 companies) as the largest sectors, underscoring their significant role in the market's response to the dual shock. Source: Refinitiv DataStream (2023) [49].

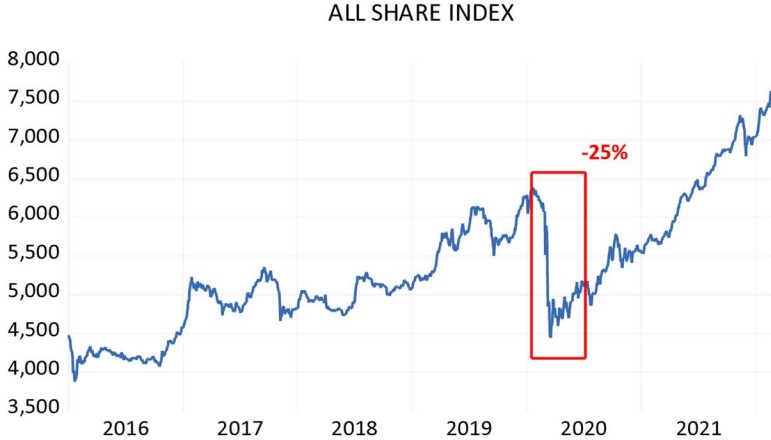

**Fig 1. All Share Index.**

Additionally, the oil and gas index (oil & gas) and the telecommunications index (telecommunications) recorded a negative 18 percent and 15 percent, and the insurance index (insurance) and the consumer goods index (consumer goods) both recorded a negative 6 percent. The technology index (technology) registered a negative 4 percent, with the lowest impacted sector being healthcare showing a negative 2 percent. The Kuwait stock market sectors reacted differently towards the pandemic, and in comparison to the Kuwait stock market index (DataStream, 2022).

### 3.2. Descriptive analysis

Sectors' returns are presented in Table 3 below. The consumer goods and technology sectors recorded losses during the analysed period as both sectors had negative means ranging between (−0.000404) and (−0.000441). On the other hand, the rest of the market sectors recorded positive returns during the analysed period as banks,

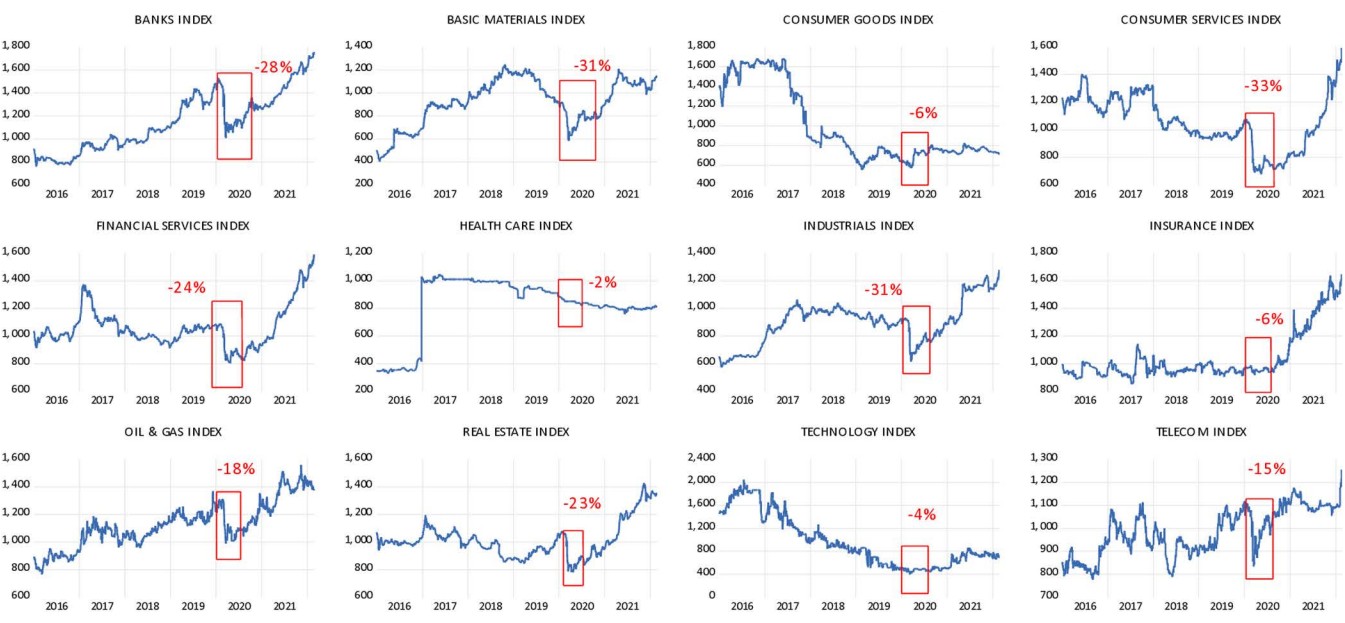

**Fig 2. Sectors.**

real estate, financial services, telecommunications, consumer services, oil & gas, healthcare, insurance, basic materials, and industrials, with positive means in the range of 0.000157, 0.000529, with the highest mean for the healthcare sector and the lowest mean for consumer services. The mean range indicates that the healthcare sector recorded the highest return profit. The standard deviation capturing volatility was lower than 5 percent, indicating significant levels of variation in the Kuwait stock market sectors during the analysed period. The standard deviation for the analysed sectors is in the range of 0.8 percent and 3.87 percent. The highest variation was identified in the healthcare sector, and the lowest in the financial services. The standard deviation results indicate that all indexes have volatilities higher than the All-share index of 0.79 percent, indicating that the sectors are more volatile than the All-share index. Hence, the skewness of five sectors is negative: consumer services, financial services, industrials, real estate, and telecommunications, ranging from (−2.195639) to (−0.209841). In addition, the series exhibited leptokurtic patterns, with values well above 3 indicating more peaks than a normal distribution with a long tail for all markets and in the range of 7.499745, 1442.357. The highest kurtosis levels were identified in the healthcare sector, and the lowest levels were in oil and gas. Last, the Jarque-Bera test for normality was significant at a 1 per cent significance level for all sectors' returns, meaning that the series were not normally distributed, which is quite common in financial time series.

The oil benchmarks returns are outlined in Table 4 below, where all market means are positive, indicating profit in returns. Moreover, the standard deviation for BRENT, DUBAI and OPEC were at 24 percent, 26 percent, and 26 percent signalling significant fluctuations during the period. The WTI recorded 32 percent, indicating that WTI was the more volatile amongst the studied oil benchmarks. The skewness of the Dubai and OPEC markets were negative (−0.963830), (−1.470631), and Brent and WTI had a positive skewness of 0.140095 and 0.760013. Furthermore, all markets were leptokurtic, indicating a more peaked distribution than a normal distribution with a long tail and in the range of 17.06003, 40.25836. The highest level of kurtosis was registered by the OPEC, and the lowest by Brent. Last, the Jarque-Bera test for normality was significant at 1 percent significance level for all oil return series, meaning the return series were not normally distributed.

**Table 3. Descriptive Statistics, Kuwait Stock Market Sectors.**

**PRICES**

| | ALL SHARE | BANKS | BASIC | CONS GOODS | CONS SER-VICES | FINAN SERVICE | HEALTH CARE | INDUST | INSURA | OIL GAS | REAL ESTATE | TECHNO | TELECOM |
|---|---|---|---|---|---|---|---|---|---|---|---|---|---|
| Mean | 5332.162 | 1147.27 | 918.7073 | 992.0754 | 1057.51 | 1060.531 | 823.3146 | 899.855 | 1039.859 | 1124.945 | 1013.579 | 964.8917 | 985.0579 |
| Std. Dev. | 808.3231 | 250.0025 | 202.05 | 373.8201 | 191.1346 | 147.0467 | 217.2546 | 157.87 | 178.3584 | 164.2419 | 127.9155 | 448.5276 | 101.9862 |
| Skew-ness | 0.63537 | 0.37606 | -0.56782 | 0.85013 | -0.00254 | 1.260968 | -1.3522 | -0.0775 | 1.841386 | 0.236556 | 1.126601 | 0.764818 | -0.11658 |
| Kurtosis | 2.953656 | 2.152754 | 2.4987 | 2.006915 | 2.36483 | 4.561583 | 3.524417 | 2.49864 | 5.230506 | 2.624017 | 4.150379 | 2.334003 | 1.971011 |
| Jarque-Bera | 108.1321 | 85.8348 | 103.0538 | 259.281 | 26.98188 | 588.4132 | 507.4986 | 18.41639 | 1239.727 | 24.42267 | 428.0194 | 186.1357 | 74.44388 |
| Proba-bility | 0.000000 | 0.000000 | 0.000000 | 0.000000 | 0.000001 | 0.000000 | 0.000000 | 0.0001 | 0.000000 | 0.000005 | 0.000000 | 0.000000 | 0.000000 |
| Obser-vations | 1605 | 1605 | 1605 | 1605 | 1605 | 1605 | 1605 | 1605 | 1605 | 1605 | 1605 | 1605 | 1605 |

**RETURNS**

| | ALL SHARE | BANKS | BASIC MATERI | CONSU GOODS | CONSU SER-VICES | FINAN SERVICE | HEALTH CARE | INDUST | INSURA | OIL GAS | REAL ESTATE | TECHNO | TELECOM |
|---|---|---|---|---|---|---|---|---|---|---|---|---|---|
| Mean | 0.000335 | 0.000407 | 0.000508 | -0.000404 | 0.000157 | 0.000266 | 0.000529 | 0.000418 | 0.000306 | 0.000273 | 0.000147 | -0.000441 | 0.00023 |
| Std. Dev. | 0.007908 | 0.009671 | 0.01304 | 0.015727 | 0.011225 | 0.007955 | 0.023273 | 0.009352 | 0.012709 | 0.013843 | 0.008444 | 0.038651 | 0.011404 |
| Skew-ness | -3.201362 | -2.568759 | 6.039736 | 0.606987 | -1.28802 | -0.87135 | 36.96876 | -2.486044 | 1.256677 | 0.342976 | -2.070539 | 0.097929 | -0.973097 |
| Kurtosis | 40.93549 | 34.68566 | 140.8841 | 31.76771 | 21.12848 | 10.53285 | 1442.357 | 40.00012 | 27.96295 | 7.499745 | 21.1549 | 15.73161 | 12.22756 |
| Jarque-Bera | 98919.76 | 68863.41 | 1280389 | 55408.52 | 22407.72 | 3995.35 | 1.39E+08 | 93147.64 | 42069.29 | 1384.669 | 23174.39 | 10835.84 | 5943.855 |
| Proba-bility | 0.000000 | 0.000000 | 0.000000 | 0.000000 | 0.000000 | 0.000000 | 0.000000 | 0.000000 | 0.000000 | 0.000000 | 0.000000 | 0.000000 | 0.000000 |

**Note:** This table reports the summary statistics of daily prices and returns for the Kuwait stock market sectors. The research sample under consideration spans December 31, 2015, to February 23, 2022— Key Takeaway: Healthcare and Basic Materials show the highest return but also extreme volatility (kurtosis>140). The Std. Dev. (standard deviation) represents the prices and returns of initial volatility patterns. The Jarque-Bera for normality is included (the p-value at 1% significance level was considered with values presented in the probability section). Source: Data Stream (2022).

**Table 4. Descriptive Statistics – Oil Benchmarks.**

**PRICES**

| | BRENT | OIL_DUBAI | OPEC | WTI |
|---|---|---|---|---|
| Mean | 59.0637 | 57.1498 | 57.09513 | 54.66583 |
| Std. Dev. | 14.12371 | 14.12773 | 15.18699 | 13.6676 |
| Skewness | −0.112905 | −0.146519 | −0.264879 | −0.179396 |
| Kurtosis | 2.652934 | 2.566099 | 2.794632 | 4.443637 |
| Jarque-Bera | 11.46536 | 18.33325 | 21.58859 | 147.9823 |
| Probability | 0.003238 | 0.000104 | 0.000021 | 0.000000 |
| Observations | 1605 | 1605 | 1605 | 1605 |

**RETURNS**

| | BRENTR | OIL_DUBAIR | OPECR | WTIR |
|---|---|---|---|---|
| Mean | 0.000958 | 0.000721 | 0.000836 | 0.001017 |
| Std. Dev. | 0.024146 | 0.025974 | 0.025918 | 0.031722 |
| Skewness | 0.140095 | −0.96383 | −1.470631 | 0.760013 |
| Kurtosis | 17.06003 | 22.41865 | 40.25836 | 28.35165 |
| Jarque-Bera | 13200.68 | 25418.38 | 93238.83 | 43054.88 |
| Probability | 0.0000000 | 0.0000000 | 0.0000000 | 0.0000000 |
| Observations | 1602 | 1602 | 1602 | 1602 |

**Note:** This table reports the summary statistics of daily prices and returns for Oil benchmarks. The research sample under consideration spans between December 31 2015, and February 23 2022. Key Takeaway: WTI is the most volatile benchmark (Std. Dev. 31.7%), while all oil benchmarks showed significant leptokurtic distribution, indicating heavy tails and extreme events. The Std. Dev. (standard deviation), represents the standard deviation of prices and returns. The Jarque-Bera for normality is included (the p-value at 1% significance level was considered with values presented in the probability section). Source: Data Stream (2022).

The correlation matrix is presented in S1 and S2 Tables illustrating the evidence of strong positive connections between six sectors of Kuwait stock markets (industrials, basic materials, banks, financial services, oil & gas and insurance), including the Kuwait All-share index and oil benchmarks (Brent, Dubai, OPEC and WTI) that is above 0.5. Basic materials and industrials exhibited the strongest correlation with all oil benchmarks. On the other hand, consumer goods and technology were negatively correlated with all oil benchmarks, and the results for all-share and Industrials had the strongest correlation with all oil benchmarks.

### 3.3. Econometric modelling

The series stationarity properties were examined through the implementation of three well-known unit root tests: The Augmented Dickey-Fuller (ADF) test [50], Phillips-Perron (PP) test [51] and Kwiatkowski-Phillips- Schmidt- Shin (KPSS) [52] to ensure that the results are robust and to ensure that there are additional tests that enable cross-checking and validating results, in line with the work done by Enders [53], Al khazali [54], Asteriou and Hall [55], Taheri [56]. Moreover, a random walk with drift was used, as the drift captures some of the market levels of variation in alignment with common trends identified in the reviewed literature, as the tests enable their implementation by considering random walk, random walk with drift, and random walk with drift and trend [57,58]. These methods highlight the primary dynamics and relationship between major oil benchmarks and Kuwait's stock market sectors, with the results of the tests being available in S3–S6 Tables.

The innovative aspect of the econometric framework lies in the integration of long- and short-run dynamics, achieved by combining Cointegration, Causality, and the Breitung and Candelon frequency domain models. This approach differs from traditional studies, which often focus on either long-term equilibrium relationships (e.g., cointegration) or short-term

interactions (e.g., Granger Causality). By employing a combination of econometric modelling processes, this study bridges the gap between these static long and short-run relationships and integrates an additional dimension that accounts for dynamic short-run interactions, providing a more comprehensive understanding of market and sectoral dynamics [17,36]. Moreover, the integration of frequency domain modelling adds additional insights as it integrates dynamic context that captures causal relationships across different time frequencies, which is particularly relevant for financial markets where relationships between variables often change over time [59]. This approach offers deeper insights into cyclical behaviour and how oil price changes affect stock market sectors differently over various time horizons [60]. The econometric modelling process section 3.2.1–3.2.5 is outlined in the Supporting Information S1 Methodology Description in S1 File.

### 3.4. Justification for model selection and robustness checks

Traditional residual-based robustness checks (e.g., autocorrelation, heteroskedasticity tests) are unsuitable for Johansen Cointegration and Granger Causality models estimated in levels, as standard assumptions do not hold. Robustness in this study is ensured by triangulating results across multiple econometric models and time horizons, consistent with best practice in financial time series econometrics [55]. Full theoretical justification is presented in detail in the Supporting Information Methodology Description in S1 File, Section 3.2.5.

## 4. Research findings

The cointegration analysis examined the association between oil benchmarks (Brent, Dubai, OPEC and WTI) and Kuwait stock market sectors, which are available in S5 Table. The results show evidence of the WTI benchmark being cointegrated with ten out of twelve Kuwait stock market sectors, including the All-share index, industrials, basic materials, banks, financial services, real estate, oil & gas, healthcare, telecommunications, insurance and technology. Moreover, the Brent oil benchmark is cointegrated with eight sectors: industrials, basic materials, healthcare, real estate, oil & gas, telecommunications, insurance and technology. The OPEC oil benchmark is cointegrated with five sectors: basic materials, healthcare, oil & gas, telecommunications, and technology. The Dubai oil benchmark is cointegrated with seven sectors: industrials, healthcare, basic materials, oil & gas, telecommunications, insurance, and technology. Furthermore, the cointegration test (S5 Table) shows that telecommunications, oil & gas, healthcare, basic materials and technology are cointegrated with all oil benchmarks.

The causality analysis between oil benchmarks (Brent, Dubai, OPEC and WTI) and Kuwait stock market sectors, including the Kuwait All-share index, offers significant evidence suggesting that the Kuwait stock market sectors exhibited a different short-term relationship towards different oil benchmarks. The results for causality are presented in S6 Table, which shows that the WTI oil benchmark is causal with five sectors (basic materials, financial services, real estate, telecommunications, and technology). The Brent oil benchmark is causal with the All-Share Index and five sectors: industrials, basic materials, banks, financial services, and technology. In addition, the OPEC oil benchmark is causal with the All-Share Index and six sectors: industrials, basic materials, banks, financial services, real estate, and oil & gas. The Dubai oil benchmark is causal with the All-Share Index and seven sectors: consumer services, industrials, basic materials, banks, financial services, real estate, and consumer goods. The causality test illustrates that the basic materials, financial services, and real estate sectors are causal, along with the four oil benchmarks.

The frequency domain causality test supported causality findings, illustrating no causality between oil benchmarks and the insurance sector and technology. Furthermore, domain causality results illustrate causality between oil benchmarks and real estate and financial services sectors. Interestingly, the research findings provide evidence of a strong correlation between oil benchmarks and the basic materials index, as well as a long-term and short-term relationship between them. Several notable studies, including those by Granger, Huangb, and Yang [61], Miller and Ratti [62] and Muhtaseb and Al-Assaf [63], have used these approaches to examine long-term relationships between oil prices and stock markets across various economies and markets, including Kuwait. These studies provide contemporary evidence of the value and

significance of these econometric models. The research findings suggest a long-run equilibrium relationship between oil prices and stock markets. This relationship has corresponding implications with Frank Knight's Risk and Uncertainty Theory [64], in which measurable risk is separated from unmeasurable uncertainty. The cointegration relationship indicates that although a few trends or patterns in oil prices could be measured and expected (risk), an unpredictable component (uncertainty) that affects the stock market sectors exists. Granger Causality testing was also applied to address linear prediction, particularly in cases where one event precedes another. It is often observed that unknown external factors influence variables. Research by Huang, Masulis, and Stoll [65] and Lee, Yang, and Huang [66] used Granger Causality to investigate the relationship between oil and stock returns. Additionally, frequency domain Causality was utilised to analyse the dynamic interactions between stock returns and Brent oil prices. The reviewed literature shows how many studies have employed the outlined techniques and reported significant findings suggesting their suitability for stock market analysis [67–70]. Understanding the long-run and short-run relationship between oil prices and stock markets allows investors to identify the risk and uncertainty components that drive market dynamics. This distinction is of fundamental importance to investors who must comply with decision-making pressures under the influence of both measurable risk and unmeasurable uncertainty.

## 5. Critical discussions

The research findings provide significant insights into the extent to which a small oil-dependent economy like Kuwait was affected by the 2020 dual shock. The research findings revealed that March 23, 2020, recorded the lowest point across markets. The consumer services (consumer services), the industrials index (industrials), and the basic materials index (basic materials) were the most affected, dropping by more than 30 per cent, which was higher than the All Share Index, which was negative 25 per cent. In addition, the results show interesting insights, as they illustrate that the banking sector has the same relationship towards oil benchmarks as the Kuwait stock market All Share Index.

Additionally, in line with early research studies examining Kuwait's stock exchange, the Kuwait economic model is significantly exposed to the oil market dynamics [6–11]. The outcomes of this research study support the evidence that the Kuwait stock market has a positive relationship with oil. This is unsurprising due to the country's heavy reliance on oil resources and its lack of economic diversification. In line with Bahrani and Filfilan [71] and Al-Refai, Zaitun & Eissa [72], the Kuwait stock market responded negatively to the dual shock. Moreover, in line with Alshihab and Alshammari [11], their findings illustrate a long-term and short-term relationship between oil and the Kuwait stock market. The findings suggest that the WTI is cointegrated with Kuwait's all-share index (long-term relationship), and Brent, OPEC, and oil Dubai are causal with Kuwait's All-share index (short-term relationship). In addition, the correlation test for the Kuwait stock market sectors shows that basic materials and industrials strongly correlate with all oil benchmarks. Further evidence is added by the outcomes of the cointegration test showing that telecommunications, oil & gas, healthcare, basic materials and technology are cointegrated with all oil benchmarks.

On the other hand, and in contrast, the research study developed by Kisswani and Elian [9] illustrates a long-run relationship between Kuwait stock market returns and both oil prices (Brent and WTI), in which the daily oil price shocks have a negative impact on stock returns. This research study illustrates that the Kuwait stock market is cointegrated with only WTI, and there is no evidence of cointegration with the Brent oil benchmark. In addition, the causality tests show that financial services, basic materials, and real estate are causal, along with all oil benchmarks. The frequency domain causality graphs (see S1–S9 Figs.) supported the Granger Causality findings, as they illustrate no causality between oil benchmarks and the insurance and technology sectors. In addition, the frequency domain causality test helps to clarify the causality between oil benchmarks and the real estate and financial services sectors. Remarkably, the study findings illustrate a strong correlation between oil benchmarks and the basic materials index, both in the long and short term. Conversely, the results show that the consumer services index had no long-term or short-term relationship with all oil benchmarks except a causal relationship with the Dubai oil benchmark. Lastly, the core research findings show that basic

materials and industrials have the strongest correlation with all oil benchmarks among all sectors, including the Kuwait All-share index. However, consumer goods and technology recorded a negative correlation with oil benchmarks. Quite interestingly, the basic materials sector has long-run and short-run dynamics with all oil benchmarks. This illustrates that the behaviour of the oil benchmark price highly influences the basic materials sector.

Furthermore, the WTI exhibited evidence of long-run and short-run dynamics towards three sectors (financial services, real estate and telecommunications). The Brent oil benchmark provides significant evidence of the existence of long-run and short-run dynamics towards the real estate sector. The OPEC oil benchmark exhibited evidence of long-run and short-run dynamics towards the oil and gas sector. The findings illustrate that Brent oil benchmark price behaviour has a major influence on real estate sector index behaviour, and OPEC oil benchmark price behaviour significantly influences the oil and gas sector index behaviour. The combination of multiple methodologies under different scenarios characterised by extensive market uncertainty is identified as vital, as it helps to shed more light on the impact of the dual market shock on Kuwait's stock market and enables cross-checking results. It is important to offer robust insights to investors who invest in countries that depend on oil, such as Kuwait, as it allows them to view the impact from multiple perspectives. Each method has its strengths, and when combined, they offer rich insights into their respective roles and purposes in financial market research.

The research findings are consistent with studies from other GCC markets. For instance, Alotaibi and Morales [1] found that March 23, 2020, recorded the lowest point across GCC markets during the 2020 Global Health Crisis. The Dubai index (DFMGI) was the most impacted, dropping by 37%, followed by the Kuwaiti index (KSE), which fell by 25%. On the other hand, the Saudi index (TASI), the Bahrain index (Bahrain), Qatar index (Qatar), and Oman (Muscut) recorded the lowest impact, with drops of 24%, 16%, 14%, and 13%, respectively [1]. On the other hand, at a sectoral level, the analyses reveal a consistent pattern: industrials, oil and gas sectors in Saudi Arabia and Qatar, while the services sector was the most impacted sector for the UAE, Oman, and Kuwait [73]. Moreover, Mhadhabi et al. [74] confirm the high sensitivity of industrials, oil and gas sectors to oil prices in Saudi Arabia and Qatar stock markets over a longer period, 2008−2022. In addition, Ben Amar et al. [75] further identify that financial services and industrials are the most impacted sectors by oil volatility for Saudi Arabia, Qatar, and Oman stock markets over the period from 2006 to 2022. In the Saudi market, specifically, Elamer et al. [76] found that during COVID-19, the industrials, basic materials, banking, oil and gas sectors were the most affected, while the real-estate, insurance, and healthcare sectors were the least affected.

The novelty and contributions of this paper fulfil three research gaps:

- There is a dearth of research studies that examine the relationship between crude oil prices and the stock market during the dual market shock. Understanding the impact of the COVID-19 pandemic and the price war on oil prices, returns, and volatility patterns provides valuable insights for investors and policy-makers regarding market sensitivity to shocks that are vital for investment and hedging activities.

- There is a lack of research studies that combine multiple traditional methodologies, such as the Johansen and Juselius Cointegration test [16], Engle and Granger Causality test [17], Granger Causality test [18], and Breitung and Candelon Frequency Domain Causality test [19], focused on Kuwait's stock market at the sectoral level. The combination of multiple methodologies enhances understanding of the dual shock's effects on Kuwait's stock market. Such insights are valuable for investors targeting oil-dependent economies like Kuwait, allowing for a multidimensional view and offering a comprehensive insight into the impact of a dual shock on Kuwait's stock market. The econometric working framework enhanced the analysis while highlighting the comparative strengths of well-established econometric models.

- There is a dearth of research studies examining the dynamic relationship between oil prices and the Kuwaiti Stock Market sectors amidst the dual shock. This paper adds value to the literature through the comprehensive analysis of Kuwaiti stock market sectors to shed light on how they responded to the combined shocks of the COVID-19 pandemic and the oil price war between Saudi Arabia and Russia.

To the best of the authors' knowledge, the 2020 dual shock remains underexplored in the academic literature, with a significant lack of studies in the context of Kuwait's stock market at a sectoral level.

## 6. Conclusion and policy implications

The Kuwait stock market sectors can be viewed as an attractive destination for investors when they develop their portfolios because they exhibit different types of behaviour when compared to each other and the Kuwait Stock Market All-share index. Therefore, understanding the long-term and short-term relationship between oil benchmarks and Kuwait's stock market sector indices can benefit portfolio management decision-making. Sectoral analysis enriches the decision-making process and supports investors in capturing potential losses and investment opportunities, monitoring their investments, and considering the importance of hedging techniques and strategies to counter and manage the effects of market uncertainty. The importance of strategic planning emerges as critical to understanding the behaviour and performance of oil prices and oil-dependent industries, investors, speculators, and other oil market participants. In turbulent times, uncertainty and business risk grow, and investors should pay more attention to business tactics and hedging strategies. Therefore, this research study contributes to a better understanding of the impact of oil prices on the Kuwait stock market sectors and offers insights to help minimising the damage and high costs associated with business environments that are affected by risk and uncertainty derived from oil shocks.

Furthermore, policy-makers should closely monitor the effects of oil price fluctuations as oil market uncertainty has significant spillover effects on the country's stock market that can spread towards the financial system and the macro economy. For instance, the relationship between oil prices and Kuwait's stock market sectors exhibits dynamic patterns in the long and short run, bringing further uncertainty to Kuwait's stock market and adding further difficulties to the government and policy-makers in the design and implementation of strategies to counteract their exposure to oil shocks. In addition, Kuwait policy-makers need to address the lack of diversification in Kuwait's economy and uplift the non-oil sectors to increase economic growth and ensure that Kuwait's exposure to the oil sector and associated risks are addressed within the context of the global sustainability goals.

## Supporting information

**S1 Table. Correlation Findings for Prices.**
(DOCX)

**S2 Table. Correlation Findings for Returns.**
(DOCX)

**S1 File. Methodology Description.**
(DOCX)

**S3 Table. Kuwait Sectors Stationarity test.**
(DOCX)

**S4 Table. Stationarity Test Oil Benchmarks.**
(DOCX)

**S5 Table. Cointegration Findings.**
(DOCX)

**S6 Table. Causality Findings.**
(DOCX)

**S1 Fig. ALL-Share Domain Causality Results.**
(TIF)

**S2 Fig. Consumer Services.**
(TIF)

**S3 Fig. Financial Services.**
(TIF)

**S4 Fig. Industrials.**
(TIF)

**S5 Fig. Insurance.**
(TIF)

**S6 Fig. Oil & Gas.**
(TIF)

**S7 Fig. Real Estate.**
(TIF)

**S8 Fig. Technology.**
(TIF)

**S9 Fig. Telecommunication.**
(TIF)

## Author contributions

**Supervision:** Lucía Morales.

**Writing – original draft:** Talal alotaibi.

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
