## [Decision Letter · Decision Letter 0]

13 Aug 2024

PONE-D-23-38848Sectoral Sensitivity of the Kuwait Stock Market to a Dual ShockPLOS ONE

Dear Dr. alotaibi,

Thank you for submitting your manuscript to PLOS ONE. After careful consideration, we feel that it has merit but does not fully meet PLOS ONE’s publication criteria as it currently stands. Therefore, we invite you to submit a revised version of the manuscript that addresses the points raised during the review process.

The manuscript has been evaluated by two reviewers, and their comments are available below.

Could you please revise the manuscript to carefully address the concerns raised?

We look forward to receiving your revised manuscript.

Kind regards,

Avanti Dey, PhD

Staff Editor

PLOS ONE

Journal Requirements:

Additional Editor Comments (if provided):

Reviewers' comments:

Reviewer's Responses to Questions

**Comments to the Author**

1. Is the manuscript technically sound, and do the data support the conclusions?

Reviewer #1: Yes

Reviewer #2: Partly

Reviewer #3: Yes

2. Has the statistical analysis been performed appropriately and rigorously? 

Reviewer #1: Yes

Reviewer #2: No

Reviewer #3: Yes

3. Have the authors made all data underlying the findings in their manuscript fully available?

Reviewer #1: Yes

Reviewer #2: Yes

Reviewer #3: Yes

4. Is the manuscript presented in an intelligible fashion and written in standard English?

Reviewer #1: Yes

Reviewer #2: Yes

Reviewer #3: Yes

5. Review Comments to the Author

Reviewer #1: This article focuses on the sectoral sensitivities of the Kuwaiti stock market in the presence of double shocks and uses models such as the dynamic causal model and the frequency domain model to determine the long-run and short-run relationship. The conclusions and discussion are interesting and the policy recommendations are relatively detailed.

Reviewer #2: Dear Authors,

Thank you for submitting your paper to PLOS One. Please refer to the attached file for detailed feedback on your paper. I hope you find the suggestions helpful for revising your manuscript.

Good luck with your revisions.

Best regards,

Reviewer #3: This is an excellent paper embodying innovation, originality, and creativity. It represents a timely piece of research on an important topical issue and an excellent command of analytical techniques.

I do not suggest any changes except for serious proofreading and minor editing.

6. PLOS authors have the option to publish the peer review history of their article (what does this mean? ). If published, this will include your full peer review and any attached files.

**Do you want your identity to be public for this peer review?** For information about this choice, including consent withdrawal, please see our Privacy Policy .

Reviewer #1: No

Reviewer #2: **Yes: ** Masud Alam

Reviewer #3: No

---

## [Author Response · Author response to Decision Letter 1]

13 Mar 2025

We would like to thank you for all the comments and feedback on our manuscript. We have completed all requested changes and have developed a comprehensive report to support our manuscript resubmission.

---

## [Decision Letter · Decision Letter 1]

2 Apr 2025

PONE-D-23-38848R1Sectoral Sensitivity of the Kuwait Stock Market to a Dual ShockPLOS ONE

Dear Dr. alotaibi,

Thank you for submitting your manuscript to PLOS ONE. After careful consideration, we feel that it has merit but does not fully meet PLOS ONE’s publication criteria as it currently stands. Therefore, we invite you to submit a revised version of the manuscript that addresses the points raised during the review process.

We look forward to receiving your revised manuscript.

Kind regards,

Jamel Boukhatem

Academic Editor

PLOS ONE

Reviewers' comments:

Reviewer's Responses to Questions

**Comments to the Author**

1. If the authors have adequately addressed your comments raised in a previous round of review and you feel that this manuscript is now acceptable for publication, you may indicate that here to bypass the “Comments to the Author” section, enter your conflict of interest statement in the “Confidential to Editor” section, and submit your "Accept" recommendation.

Reviewer #2: All comments have been addressed

Reviewer #3: All comments have been addressed

2. Is the manuscript technically sound, and do the data support the conclusions?

Reviewer #2: Partly

Reviewer #3: Yes

3. Has the statistical analysis been performed appropriately and rigorously? 

Reviewer #2: No

Reviewer #3: Yes

4. Have the authors made all data underlying the findings in their manuscript fully available?

Reviewer #2: No

Reviewer #3: Yes

5. Is the manuscript presented in an intelligible fashion and written in standard English?

Reviewer #2: Yes

Reviewer #3: Yes

6. Review Comments to the Author

Reviewer #2: Please see the attached comments and I request you carefully consider the comments and address them in your revised paper. Best,

Reviewer #3: The abstract is too long. It should be more succinct and precise. Currently, it looks like an executive summary.

7. PLOS authors have the option to publish the peer review history of their article (what does this mean? ). If published, this will include your full peer review and any attached files.

**Do you want your identity to be public for this peer review?** For information about this choice, including consent withdrawal, please see our Privacy Policy .

Reviewer #2: **Yes: ** Masud Alam

Reviewer #3: No

---

## [Author Response · Author response to Decision Letter 2]

25 Jun 2025

We have completed all requested changes and have developed a comprehensive report to support our manuscript resubmission.

The core points to consider include:

1. Updated manuscript with track changes to ensure that you are able to identify all amendments and additional information.

2. Updated manuscript without track changes.

3. Response letter with all comments addressed and full explanations provided that you can find in our submission.

We remain at your disposal for any further comments or clarifications that might be needed.

Yours sincerely,

Talal Alotaibi

---

## [Decision Letter · Decision Letter 2]

16 Jul 2025

PONE-D-23-38848R2Sectoral Sensitivity of the Kuwait Stock Market to a Dual ShockPLOS ONE

Dear Dr. Morales,

Thank you for submitting your manuscript to PLOS ONE. After careful consideration, we feel that it has merit but does not fully meet PLOS ONE’s publication criteria as it currently stands. Therefore, we invite you to submit a revised version of the manuscript that addresses the points raised during the review process.

We look forward to receiving your revised manuscript.

Kind regards,

Jamel Boukhatem

Academic Editor

PLOS ONE

Journal Requirements:

Reviewers' comments:

Reviewer's Responses to Questions

**Comments to the Author**

1. If the authors have adequately addressed your comments raised in a previous round of review and you feel that this manuscript is now acceptable for publication, you may indicate that here to bypass the “Comments to the Author” section, enter your conflict of interest statement in the “Confidential to Editor” section, and submit your "Accept" recommendation.

Reviewer #4: (No Response)

2. Is the manuscript technically sound, and do the data support the conclusions?

Reviewer #4: Yes

3. Has the statistical analysis been performed appropriately and rigorously? 

Reviewer #4: Yes

4. Have the authors made all data underlying the findings in their manuscript fully available?

Reviewer #4: Yes

5. Is the manuscript presented in an intelligible fashion and written in standard English?

Reviewer #4: Yes

6. Review Comments to the Author

Reviewer #4: Revisions Required

(1). Redundancy: Reduce repetition across Introduction, Discussion, and Abstract (e.g., repeated reporting of sectoral drops). The formatting needs to be improved, especially for sector drop figures, to ensure consistency with axis scales, labelling, and legends. Also, the author should summarize key takeaways below descriptive tables to aid interpretability.

(2). Optionally, compare findings with other GCC economies to enhance regional relevance.

(3). Although the rationale for avoiding robustness checks on residuals is justified, a clearer explanation (or relegation to the Appendix) would help clarify.

(4). Address minor typographical issues (e.g., inconsistent spacing, variable formatting) and revise for brevity where sections become overly lengthy.

7. PLOS authors have the option to publish the peer review history of their article (what does this mean? ). If published, this will include your full peer review and any attached files.

**Do you want your identity to be public for this peer review?** For information about this choice, including consent withdrawal, please see our Privacy Policy .

Reviewer #4: **Yes: ** Semiu Ayinla Alayande

---

## [Author Response · Author response to Decision Letter 3]

13 Aug 2025

Dear Reviewers,

Thank you very much for all your suggestions. All issues were addressed as follows.

Editor/s and Reviewer/s Feedback

This manuscript presents a comprehensive and timely examination of Kuwait's stock market sectoral response to the dual shock of the COVID-19 pandemic and the 2020 oil price war. The study tends to be original and methodologically sound. It will also be highly relevant to academic researchers and policy-makers interested in the dynamics of oil-dependent economies. The paper addresses a clear gap in the literature, providing a sectoral-level analysis of an underexplored context (Kuwait) about a significant dual economic health crisis. Using Johansen Cointegration, Granger Causality, and Frequency Domain Causality provides a robust multi-angle approach. Strong empirical analysis highlighting differential sector responses (e.g., Basic Materials, Industrials, Consumer Services) led to the effective use of figures and tables to visualize market drops. Also, the findings will be helpful for investment strategy and policy design, especially regarding risk management and economic diversification in oil-dependent economies. However, some clarity still needs to be sorted out.

Revisions Required

(1). Redundancy: Reduce repetition across Introduction, Discussion, and Abstract (e.g., repeated reporting of sectoral drops). The formatting needs to be improved, especially for sector drop figures, to ensure consistency with axis scales, labelling, and legends. Also, the author should summarize key takeaways below descriptive tables to aid interpretability.

Response:

Thank you for pointing this out. We have gone through the Abstract, Introduction, and Discussion sections to remove repeated reporting of sectoral drops, keeping each section focused and avoiding duplication.

To make the tables easier to interpret, we have added short “Key Takeaways” summaries directly beneath each descriptive table as follows:

Table 2

Table 2: This details the Kuwait All Share index and its sectors, with Financial Services (49 companies) and Real Estate (39 companies) as the largest sectors, underscoring their significant role in the market’s response to the dual shock. Source: Refinitiv DataStream (2023).

Figure 2

Figure 2: Kuwait Stock Market Sectors Decline amidst the COVID-19 Pandemic. The figure highlights the lowest point reached by the sectors, which was recorded on March 23, 2020. Sectors such as Consumer Services, Industrials, and Basic Materials declined by more than 30%, while Oil and Gas and Telecommunications decreased by 18% and 15% respectively. Insurance and Consumer Goods dropped by 6%, and Technology and Healthcare dropped by 4% and 2%, respectively. Source: DataStream (2023).

Table 3

Note: This table reports the summary statistics of daily prices and returns for Kuwait stock market sectors. The research sample under consideration spans December 31, 2015, to February 23, 2022— Key Takeaway: Healthcare and Basic Materials show the highest return but also extreme volatility (kurtosis>140). The Std. Dev. (standard deviation) represents the prices and returns initial volatility perfomance. The Jarque-Bera for normality is included (the p-value at 1% significance level was considered with values presented in the probability section). Source: Data Stream (2022)

Table 4

Note: This table reports the summary statistics of daily prices and returns for Oil benchmarks. The research sample under consideration spans between December 31 2015 and February 23 2022. Key Takeaway: WTI is the most volatile benchmark (Std. Dev. 31.7%), while all oil benchmarks showed significant leptokurtic distribution, indicating heavy tails and extreme events. The Std. Dev. (standard deviation) representing initial volatility patterns. The Jarque-Bera for normality is included (the p-value at 1% significance level was considered with values presented in the probability section). Source: Data Stream (2022).

(2). Optionally, compare findings with other GCC economies to enhance regional relevance.

We appreciate this suggestion and agree it adds value. We have now included a comparative discussion in the Discussion section, drawing on recent studies of other GCC stock markets during the COVID-19 and oil price shocks. This comparison shows common patterns, such as the energy and industrial sectors being the most affected, as follows:

These findings are consistent with studies from other GCC markets. For instance, Alotaibi and Morales (2022) found that March 23, 2020, recorded the lowest point across GCC markets during the global health crisis. The Dubai index (DFMGI) was the most impacted, dropping by 37%, followed by the Kuwaiti index (KSE), which fell by 25%. On the other hand, the Saudi index (TASI), the Bahrain index (Bahrain), Qatar index (Qatar), and Oman (Muscut) recorded the lowest impact, with drops of 24%, 16%, 14%, and 13%, respectively (Alotaibi and Morales, 2022). On the other hand, at a sectoral level, the analyses reveal a consistent pattern: Industrials and Oil and Gas sectors in Saudi Arabia and Qatar, while the Services sector was the most impacted sector for the UAE, Oman, and Kuwait (Alsamman and Akkas, 2022). Moreover, Mhadhabi et al. (2024) confirm the high sensitivity of Industrials and Oil and Gas sectors to oil prices in Saudi Arabia and Qatar stock markets over a longer period, 2008-2022. In addition, Ben Amar et al. (2025) further identify that Financial Services and Industrials are the most impacted sectors by oil volatility for Saudi Arabia, Qatar, and Oman stock markets over the period from 2006 to 2022. In the Saudi market, specifically, Elamer et al. (2022) found that during COVID-19, the Industrials, Basic Materials, Banking, and oil and Gas sectors were the most affected, while the Real-estate, Insurance, and Healthcare sectors were the least affected.

(3). Although the rationale for avoiding robustness checks on residuals is justified, a clearer explanation (or relegation to the Appendix) would help clarify.

Response:

We have clarified our reasoning for not conducting robustness checks on residuals in the Methodology section, making the explanation more concise and direct. For readers who want full details, The original section was moved to the appendix, and the following paragraph was added to the text.

3.2.5 Justification for Model Selection and Robustness Checks

Traditional residual-based robustness checks (e.g. autocorrelation, heteroskedasticity tests) are unsuitable for Johansen Cointegration and Granger Causality models estimated in levels, as standard assumptions do not hold. Robustness in this study is ensured by triangulating results across multiple econometric models and time horizons, consistent with best practice in financial time series econometrics (Asteriou & Hall, 2015). Full theoretical justification is detailed in Appendix 2 Section 3.2.5.

(4). Address minor typographical issues (e.g., inconsistent spacing, variable formatting) and revise for brevity where sections become overly lengthy.

Response:

We have proofread the manuscript thoroughly to correct spacing, punctuation, and formatting inconsistencies. We also shortened some longer sections in the Introduction and Discussion, so they read more concisely while keeping all key points intact.

---

## [Editor Report · Decision Letter 3]

15 Aug 2025

Sectoral Sensitivity of the Kuwait Stock Market to a Dual Shock

PONE-D-23-38848R3

Dear Dr. Morales,

We’re pleased to inform you that your manuscript has been judged scientifically suitable for publication and will be formally accepted for publication once it meets all outstanding technical requirements.

Kind regards,

Jamel Boukhatem

Academic Editor

PLOS ONE
---

## [Editor Report · Acceptance letter]

PONE-D-23-38848R3

PLOS ONE

Dear Dr. Morales,

I'm pleased to inform you that your manuscript has been deemed suitable for publication in PLOS ONE. Congratulations! Your manuscript is now being handed over to our production team.

Kind regards,

on behalf of

Professor Jamel Boukhatem

Academic Editor

PLOS ONE